# QUANTIZE-THEN-RECTIFY: ACCELERATING VQ-VAE TRAINING IN LATENT FEATURE SPACE

## ABSTRACT

Visual tokenizers are pivotal in multimodal large models, acting as bridges between continuous inputs and discrete token. Nevertheless, training high-compression-rate VQ-VAEs remains computationally demanding, often necessitating thousands of GPU hours. This work demonstrates that a pre-trained VAE can be efficiently transformed into a VQ-VAE by controlling quantization noise within the VAE's tolerance threshold. We present **Quantize-then-Rectify (ReVQ)**, a framework leveraging pre-trained VAEs to enable rapid VQ-VAE training with minimal computational overhead. By integrating **channel split quantization** to enhance codebook capacity and a **post rectifier** to mitigate quantization errors, ReVQ compresses ImageNet images into at most 512 tokens while sustaining competitive reconstruction quality (rFID = 0.82). Significantly, ReVQ reduces training costs by over two orders of magnitude relative to state-of-the-art approaches: ReVQ finishes full training on a single NVIDIA 4090 in approximately 22 hours, whereas comparable methods require 4.5 days on a 32 A100 GPUs. Experimental results show that ReVQ achieves superior efficiency-reconstruction trade-offs.

## 1 INTRODUCTION

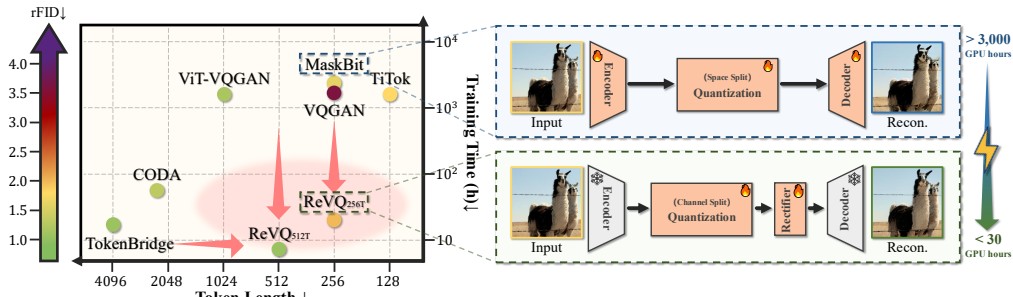

Figure 1: ReVQ achieves the optimal trade-off between training efficiency (requiring only 1 day on a single NVIDIA 4090) and compression ratio ($\leq 512$ tokens for $256 \times 256$ images), maintaining highly competitive reconstruction quality, as demonstrated by an rFID score of $0.82$, when compared to state-of-the-art VQ-VAEs like MaskBit (Weber et al., 2024) (4.5 day on 32 A100s).

Large language models (LLMs) (Brown, 2020) have revolutionized artificial intelligence by utilizing discrete token sequences for next-token prediction. For integrating vision with LLMs, visual tokenizers play a critical role in bridging continuous image spaces and discrete input formats of LLMs. Vector-quantized variational autoencoders (VQ-VAEs) (Van Den Oord et al., 2017) serve as foundational components by discretizing image latent spaces, enabling alignment between visual and linguistic modalities in vision-LLM architectures (Esser et al., 2021; Razavi et al., 2019).

Despite significant advancements in reconstruction quality (Yu et al., 2022; Chang et al., 2022), modern VQ-VAEs still face a fundamental challenge: a trade-off between **training efficiency** and **compression ratio**. Current approaches can be broadly categorized into two distinct categories (Fig. 1). (1) **high-compression but high-cost methods** (e.g., MaskBit (Weber et al., 2024), $\leq 256$ tokens) demand substantial computational resources, requiring over $3,000$ gpu hours on A100 clusters. This high computational cost limits accessibility to well-resourced institutions. (2) **efficient but**

**low-compression methods** (e.g., TokenBridge (Wang et al., 2025), 4096 tokens; CODA (Liu et al., 2025), 2560 tokens) leverage pre-trained VAEs for rapid quantization but fail to achieve the short token lengths necessary for downstream generative tasks (Rombach et al., 2022).

This work addresses the need for a VQ-VAE framework that achieves high compression ratios and efficient training. We uncover an inherent relationship between VAEs and VQ-VAEs: under specific conditions, a pre-trained VAE can be systematically transformed into a VQ-VAE with minimal computational overhead. Unlike previous attempts such as TokenBridge (Wang et al., 2025) and CODA (Liu et al., 2025), which compromise on token length, our **Quantize-then-Rectify (ReVQ)** framework leverages pre-trained VAEs to facilitate fast VQ-VAE training while maintaining high compression performance (Fig. 2). By integrating **channel split quantization** to enhance codebook capacity and a **post rectifier** to alleviate quantization errors, ReVQ compresses ImageNet images into at most $512$ tokens while sustaining competitive reconstruction quality (rFID = $0.82$). ReVQ completes full training on a single NVIDIA 4090 in approximately 22 hours, in contrast to comparable methods that require 4.5 days on a 32 A100 GPUs. The core contributions of this work are as follows:

- **Connection between VAE and VQ-VAE:** We formalize the boundary conditions for converting a VAE into a VQ-VAE, establishing a linkage between these two model classes.

- **Representative quantizer design:** A post rectifier is introduced to alleviate quantization errors, optimized for reduced gradient noise during training.

- **Efficient ReVQ framework:** ReVQ transforms a VAE into a VQ-VAE within one day on a single NVIDIA 4090, trained solely with an $l_2$ loss, achieving competitive reconstruction quality while delivering a two-order-of-magnitude improvement in training speed.

- **Extensive experimental analysis:** Results on ImageNet demonstrate ReVQ achieves superior balance between training efficiency and compression ratio, encoding images into $\leq 512$ tokens with competitive rFID while drastically reducing computational demands.

## 2 RELATED WORK

VQ-VAEs (Van Den Oord et al., 2017) serve as a highly effective bridge between continuous and discrete spaces, enabling the application of deep learning in diverse domains such as image understanding (Bao et al., 2022; Ge et al., 2024; Jin et al., 2024) and generation (Esser et al., 2021; Chang et al., 2022; Tian et al., 2024). Existing efforts to improve VQ-VAEs can be broadly categorized into two main approaches: **model structure** and **quantization strategy**.

**Model Structure.** The original VQ-VAE (Van Den Oord et al., 2017) first introduced an effective framework for discretizing continuous data. However, early VQ-VAEs often suffered from suboptimal reconstruction quality. Subsequent research focused on refining model architectures to address this limitation. First, diverse backbone networks were developed to enhance model capacity. VQ-VAE2 (Razavi et al., 2019) employed a multi-scale quantization strategy to preserve high-frequency details, while integrating Vision Transformers (Yu et al., 2022; 2024b; Cao et al., 2023) significantly improved representational power. Second, the incorporation of generative adversarial networks (GANs) (Goodfellow et al., 2014) brought substantial advancements. VQGAN (Esser et al., 2021) improved the perceptual quality of reconstructed images by combining GANs with perceptual loss functions (Larsen et al., 2016; Johnson et al., 2016). Third, semantic supervision emerged as an effective approach. VAR (Tian et al., 2024) utilized DINO (Oquab et al., 2023) as a semantic prior to enhance reconstruction fidelity, while ImageFolder (Li et al., 2025) introduced a semantic branch in the quantization module supervised by contrastive loss.

**Quantization Strategy.** Conventional VQ-VAEs rely on nearest-neighbor search to map features to codebook entries, a method that has been shown to have limitations in optimization stability and codebook utilization. For improving optimization stability, various techniques have been proposed to enhance training robustness: low-dimensional codebooks (Yu et al., 2022), shared affine transformations (Huh et al., 2023; Zhu et al., 2024), specialized initializations (Huh et al., 2023; Zhu et al., 2024), and model distillation (Yu et al., 2024b). ViT-VQGAN (Yu et al., 2022) observed the sparsity in high-dimensional feature spaces and demonstrated that reducing codebook dimensionality increases feature-code proximity, thereby improving code utilization. Shared affine transformations (Huh et al.,

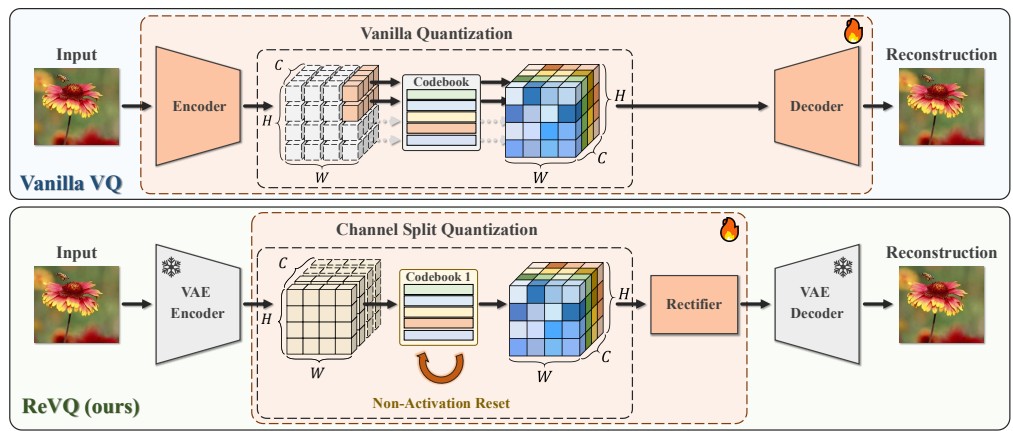

Figure 2: Comparison of Vanilla VQ and ReVQ. (Top) Vanilla VQ trains encoder, decoder, and quantizer from scratch, demanding substantial computational resources. (Bottom) ReVQ freezes pre-trained VAE encoder/decoder parameters, training only a quantizer and lightweight rectifier for high-performance VQ-VAE. To boost quantizer capacity, it uses channel split quantization and ensures codebook utilization via non-activation reset.

2023; Zhu et al., 2024) highlighted the sparsity and slowness of conventional codebook updates, proposing an affine layer to convert sparse updates into dense transformations for more efficient adaptation. K-Means initialization (Huh et al., 2023; Zhu et al., 2024) was identified as a reliable method to mitigate premature convergence from arbitrary initializations, while distillation from pre-trained models like MaskGiT (Chang et al., 2022) further boosted performance (Yu et al., 2024b). OptVQ (Zhang et al., 2024) applied optimal transport theory to model the global distributional relationship between codes and features, enhancing matching accuracy. To expand codebook capacity, strategies such as residual mechanisms (Lee et al., 2022), multi-head mechanisms (Zheng et al., 2022), and multi-group quantization (Ma et al., 2025) have been proposed. Meanwhile, lookup-free approaches like FSQ (Mentzer et al., 2023) and LFQ (Yu et al., 2024a) aimed to enhance reconstruction efficiency by avoiding explicit codebook lookups.

## 3 METHOD

Training a VQ-VAE from scratch is computationally expensive. MaskBit (Weber et al., 2024) reports 3456 GPU hours (4.5 days on a 32 A100 GPUs for 1.35M iterations) for ImageNet (Deng et al., 2009), which is prohibitive for most researchers. This work addresses the high training cost by analyzing bottlenecks in VQ-VAEs and exploring strategies to accelerate VQ-VAE training. In Section 3.1, we dissect the core components of VQ-VAE and identify time-consuming modules. Section 3.2 discusses key adaptations for converting VAEs to VQ-VAEs, including channel split quantization and non-activation reset. Finally, Section 3.3 introduces ReVQ, a quantize-then-rectify approach that transforms pre-trained VAEs into VQ-VAEs with minimal computational overhead.

### 3.1 PRELIMINARY: TIME-CONSUMING VQ-VAE TRAINING

Compact latent space representation of high-dimensional data is fundamental. Autoencoders (Hinton and Salakhutdinov, 2006) initiated the exploration of low-dimensional image encoding, while VAEs (Kingma, 2013) advanced this by introducing prior distributions, enabling data generation via latent space sampling. With the GPT era, discrete image representations became necessary to align with discrete base LLMs. VQ-VAEs (Van Den Oord et al., 2017) replaced continuous priors with discrete codebooks, gaining wide use in image generation (Esser et al., 2021; Chang et al., 2022; Rombach et al., 2022) and large-scale pre-training (Bao et al., 2022; Bai et al., 2024). Let $\mathcal{X} = \{\boldsymbol{x}_i\}_{i=1}^N$ denote the image dataset. A standard VQ-VAE consists of an encoder $f_e(\cdot)$, a decoder $f_d(\cdot)$, and a quantizer $q(\cdot)$. The encoder maps input image $\boldsymbol{x}$ to a 3D latent feature $\boldsymbol{Z}_e = f(\boldsymbol{x}) \in \mathbb{R}^{H \times W \times D}$. For each vector $\boldsymbol{z}_e \in \mathbb{R}^D$ in $\boldsymbol{Z}_e$, the quantizer finds the nearest code vector in codebook $\mathcal{C} = \{\boldsymbol{c}_1, \boldsymbol{c}_2, \ldots, \boldsymbol{c}_n\}$ via nearest-neighbor search, yielding the quantized vector $\boldsymbol{z}_q = q(\boldsymbol{z}_e)$. These form the quantized feature map $\boldsymbol{Z}_q$, from which the decoder reconstructs the image as $\hat{\boldsymbol{x}} = f_d(\boldsymbol{Z}_q)$.

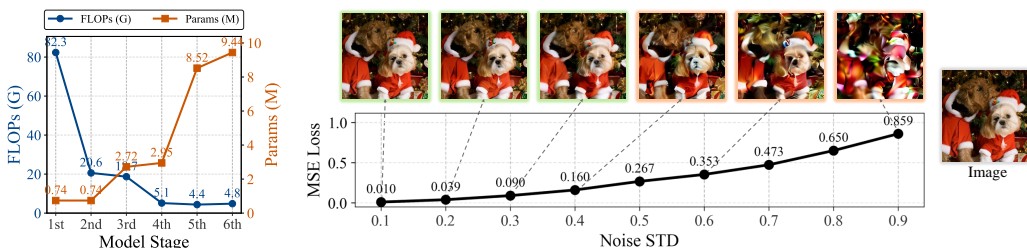

(a) Computation Statistics.     (b) Anlysis of Noise tolerant ability of the VAE under different noise levels.

Figure 3: Research Motivation. (a) Computational statistics reveal that shallow layers dominate computations, enabling substantial savings via pre-trained VAE. (b) VAE noise tolerance is demonstrated, showing conversion to VQ-VAE feasible when quantization error is below the threshold.

**Training Pipeline.** The quantizer is typically implemented via nearest neighbor search (Van Den Oord et al., 2017; Esser et al., 2021) as:

$$\boldsymbol{z}_q = q(\boldsymbol{z}_e, \mathcal{C}) = \boldsymbol{c}_k, \text{ where } k = \arg\min_j \|\boldsymbol{z}_e - \boldsymbol{c}_j\|, \tag{1}$$

with $\|\cdot\|$ denoting a distance metric (e.g., Euclidean). Large codebooks incur significant computational costs for distance matrix computation. Lookup-free quantizers (Mentzer et al., 2023; Yu et al., 2024a) avoid this by directly rounding feature map elements to integers for codebook indices. However, quantization operation may get trapped in local minima, causing "index collapse" (Huh et al., 2023). To address this, some replace nearest neighbor search with distribution matching (Zhang et al., 2024) to ensure full codebook utilization. Existing works train VQ-VAEs end-to-end by minimizing reconstruction loss $\mathcal{L}_{\text{rec}} = \|\boldsymbol{x} - \hat{\boldsymbol{x}}\|$ (Van Den Oord et al., 2017). Since reconstruction may overemphasize low-level details, perceptual and adversarial losses (Esser et al., 2021; Chang et al., 2022; Yu et al., 2022; Cao et al., 2023) are often added to enhance visual quality. Adversarial loss has the most significant aesthetic impact, followed by perceptual loss, $l_1$-based, and $l_2$-based reconstruction losses. The non-differentiable nearest neighbor search requires gradient approximation via the straight-through estimator (Bengio et al., 2013; Huh et al., 2023).

**Computation Statistics.** To understand why VQ-VAE training is time-consuming, we analyze FLOPs and parameters of a typical model (Weber et al., 2024) (see Fig. 3a). High computational burden concentrates in shallow layers due to large input resolution, while deep layers have more parameters. This motivates training VQ-VAEs on pre-downscaled inputs using pre-trained VAEs to compress images into latent spaces first. Recent works like TokenBridge (Wang et al., 2025) (4096 tokens per image) and CODA (Liu et al., 2025) (2560 tokens per image via residual coding) demonstrate fast VQ-VAE development with pre-trained VAEs, though achieving lower compression ratios than conventional models (256 tokens per image). Our work explores whether fine-tuning pre-trained VAEs can yield VQ-VAEs with comparable compression efficiency and fast training.

### 3.2 CONVERSION OF VAE INTO VQ-VAE

In this section, we observe strong noise tolerance in autoencoders and present techniques for converting a standard VAE to a VQ-VAE. We encode an ImageNet image using DC-AE (Chen et al., 2024) into a 2048 dimensional latent vector, normalize it via dataset statistics, and add Gaussian noise with varying variances before reconstruction (Fig. 3b). Results show that reconstructed images retain high quality when noise variance $\leq 0.3$ (green boxes), but degrade significantly beyond this threshold (red boxes). This indicates that while Eq. (1)-based vector quantization introduces noise, acceptable reconstruction is maintained if quantization noise stays within the VAE's tolerance threshold. Critical factors include **codebook capacity** and **optimization stability** to avoid local minima.

#### 3.2.1 CODEBOOK CAPACITY: CHANNEL SPLIT QUANTIZATION

The effective codebook capacity is critical for low quantization error. Consider a sample encoded with $B$ tokens from a codebook of size $N$. The number of token combinations is $M = N^B$, defining the upper bound of samples the codebook can represent. In practice, VQ-VAE's effective codebook

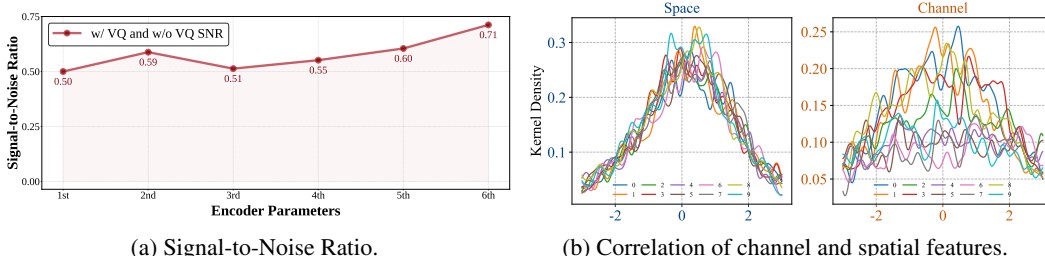

(a) Signal-to-Noise Ratio.    (b) Correlation of channel and spatial features.

Figure 4: (a) SNR of the encoder gradient norm (w/ and w/o quantization). (b) Shows feature correlations under different spliting, with channel-based spliting yielding more independent distributions.

capacity $M$ often far exceeds training data size. For instance, ImageNet has $A = 1, 281, 167 \approx 2^{20.29}$ images, and a VQ-VAE with $B = 256$ tokens and $N = 1024$ yields $M = 1024^{256} \approx 2^{2560}$, where $M \gg A$. This raises the question: why doesn't such vast capacity improve VQ-VAE training? The answer partially stems from conventional quantizers with spatial dimension spliting. To address this issue, we propose a **channel split** strategy: define $z_e^i = [\boldsymbol{Z}_e']_{(i,\cdot)}$ and apply quantization. Detailed explanations can be found in the supplementary material Appendix C.

### 3.2.2 Optimization Stability: Non-Activation Reset

Extensive research has shown that optimizing VQ-VAE is a highly non-convex problem, extremely prone to index collapse (Huh et al., 2023; Zhang et al., 2024; Zhu et al., 2024). Failing to address its unstable optimization may render increased codebook capacity ineffective. Some studies suggest that alternating K-means initialization can resolve the unstable optimization (Huh et al., 2023). Nevertheless, in large-scale problems, the overhead of K-means initialization is relatively high. OptVQ (Zhang et al., 2024) points out the fundamental reason why nearest-neighbor based quantizers are prone to local optima. Once a code $c_i$ is not selected by any sample in an iteration, it is highly likely to never be selected thereafter. Inspired by this, we identify that the key to solving this problem lies in "discovering non-activations and resetting them", thus proposing the **Non-Activation Reset** strategy. Specifically, during each training epoch, for the codebook $\mathcal{C}$, we count the activation time $t_i$ of each code $c_i$. At the end of the epoch, we sort the indices of the $N$ codes in ascending order of their $t_i$ values, obtaining $\mathbb{I} = \{i_1, i_2, \cdots, i_N\}$. When there are $r$ unactivated codes (i.e., the first $r$ indices in $\mathbb{I}$ have $t_i = 0$), we perform the following reset operation:

$$\boldsymbol{c}_{i_u} \leftarrow \boldsymbol{c}_{i_{N+1-u}} + \epsilon, \ u = 1, \cdots, r, \tag{2}$$

where $\epsilon$ is a small random perturbation to avoid overlapping between codes after reset. This operation intuitively resets unactivated points to the vicinity of highly activated codes, sharing the burden of frequently activated codes and promoting a more uniform activation frequency across codes.

We find that methods balancing codebook activation frequencies effectively prevent codebook collapse. This is also reflected in the entropy regularization proposed in LFQ (Yu et al., 2024a) and the optimal transport search in OptVQ (Zhang et al., 2024). The distinction lies in that our reset strategy requires no additional loss functions or computational steps during training. Only a single reset operation at the end of each epoch, making it a plug-and-play module in code implementation. We briefly analyze in the supplementary material (Appendix D) whether the reset strategy can reduce quantization errors.

### 3.3 REVQ: Quantize-then-Rectify

We primarily focus on analyzing the essential components for converting a VAE into a VQ-VAE purely from the quantizer perspective. However, as shown in Fig. 10, the VQ-VAE converted from DC-AE (Chen et al., 2024) can at most compress images into 512 tokens to achieve a moderately effective model if relying solely on the quantizer. Further increasing the compression ratio would lead to an exponential explosion in the required number of codebook. To address this, we introduce the **Quantize-then-Rectify (ReVQ)** framework in this section. The proposed method posits that for the quantized features $\boldsymbol{Z}_q$ from quantizer $q$, a rectifier $g$ should be constructed. The reconstructed quantized features via the ReVQ method are thus given by:

$$\boldsymbol{Z}_e' = g\left(q\left(\boldsymbol{Z}_e, \mathcal{C}\right)\right). \tag{3}$$

Table 1: Quantitative comparison with state-of-the-art methods on ImageNet.

| Type | Method | Token Length | #Codebook | SSIM↑ | PSNR↑ | LPIPS↓ | rFID↓ |
|------|--------|--------------|-----------|-------|-------|--------|-------|
| From Scratch | ViT-VQGAN | 1024 (16×16) | 8,192 | - | - | - | 1.28 |
| | Mo-VQGAN | 1024 (16×16×4) | 1,024 | 0.673 | 22.420 | 0.113 | 1.12 |
| | ImageFolder | 572 (286×2) | 4,096 | - | - | - | 0.80 |
| | VQGAN | 256 (16×16) | 16,384 | 0.542 | 19.930 | 0.177 | 3.64 |
| | MaskGIT | 256 (16×16) | 1,024 | - | - | - | 2.28 |
| | RQ-VAE | 256 (8×8×4) | 16,384 | - | - | - | 3.20 |
| | MaskBit | 256 (16×16) | 4,096 | - | - | - | 1.61 |
| | COSMOS | 256 (16×16) | 64,000 | 0.518 | 20.490 | - | 2.52 |
| | VQGAN-LC | 256 (16×16) | 100,000 | 0.589 | 23.800 | 0.120 | 2.62 |
| | LlamaGen-L | 256 (16×16) | 16,384 | 0.675 | 20.790 | - | 2.19 |
| Fine Tuning | TiTok-S-128 | 128 | 4,096 | - | - | - | 1.71 |
| | CODA | 2560 (256×10) | 65,536 | 0.602 | 22.200 | - | 1.34 |
| Frozen | TokenBridge | 4096 | 64 | - | - | - | 1.11 |
| | **ReVQ$_{512T}$** | **512** | **16,384** | **0.692** | **23.923** | **0.093** | **0.82** |
| | **ReVQ$_{256T}$** | **256** | **65,536** | **0.624** | **21.773** | **0.131** | **2.56** |
| | **ReVQ$_{256T}$** | **256** | **262,144** | **0.640** | **22.267** | **0.122** | **1.92** |

Since the rectifier $g$ is trained under relatively low-resolution cases, the comparisons in Table 2 show that ReVQ can convert a VAE into a VQ-VAE extremely efficiently on a single RTX 4090 GPU. In contrast, traditional VQ-VAE training may require 4.5 days on 32 A100 GPUs as reported in (Weber et al., 2024). We now elaborate on the rectifier model design and training loss function.

**Optimization-Friendly Rectifier Design.** ReVQ adopts a decoder-only framework, mitigating the optimization challenges of quantization noise. In contrast to VAE, quantization in VQ-VAE introduces noise that affects the encoder's gradient, hindering convergence. As shown in Fig. 4a, with identical encoder-decoder structures and the same sample, quantization leads to an increase of at least $50\%$ in the encoder's noise gradient, which impedes optimization. To address this, ReVQ employs a rectifier model $g$, a decoder-only structure that prevents quantization's impact on the encoder's gradient. In particular, we utilize the EfficientViT block (Cai et al., 2023) as the rectifier model, which avoids upsampling/downsampling of latent variables and maintains consistent dimensions. This design is inspired by DC-AE (Chen et al., 2024), which proposes a high-compression VAE architecture capable of compressing images into $2048$-dimensional vectors using a residual structure for image reconstruction. Leveraging the EfficientViT block, ReVQ framework enhances the efficiency and effectiveness of the rectifier model, optimizing training performance and compression efficiency.

**Training Loss.** Conventional VQ-VAE training typically combines loss functions like perceptual loss (Johnson et al., 2016), Patch GAN loss (Isola et al., 2017), and $l_2$/$l_1$ losses. In contrast, our ReVQ framework simplifies the training process by treating the VAE as a black box and excluding the need to compute its gradients, thereby reducing computational overhead. As a result, we exclusively apply $l_2$ loss in the latent space of $\boldsymbol{Z}_e$. The final optimization objective becomes:

$$\min_{\theta_g, \mathcal{C}} L_{\text{ReVQ}} = \|\boldsymbol{Z}_e - g\left(q(\boldsymbol{Z}_e)\right)\|_2^2, \tag{4}$$

where $\theta_g$ denotes the parameters of the rectifier model and $\mathcal{C}$ represents the codebook parameters. The detailed training algorithm for ReVQ is provided in Algorithm 1.

## 4 EXPERIMENT

In this section, we illustrate the reconstruction performance and training efficiency of ReVQ. We first detail the experimental setup in Section 4.1, followed by presenting quantitative and qualitative comparisons between ReVQ and other VQ-VAE methods in Section 4.2. Subsequently, ablation experiments are conducted in Section 4.4 and Section 4.5 to validate the effectiveness of the quantization module and the rectification module, respectively.

### 4.1 EXPERIMENTAL DETAIL

**Model Setting.** Our model consists of a quantizer and a decoder. Building upon the continuous latent space of the VAE, we utilize and freeze both the encoder and decoder weights from the DC-

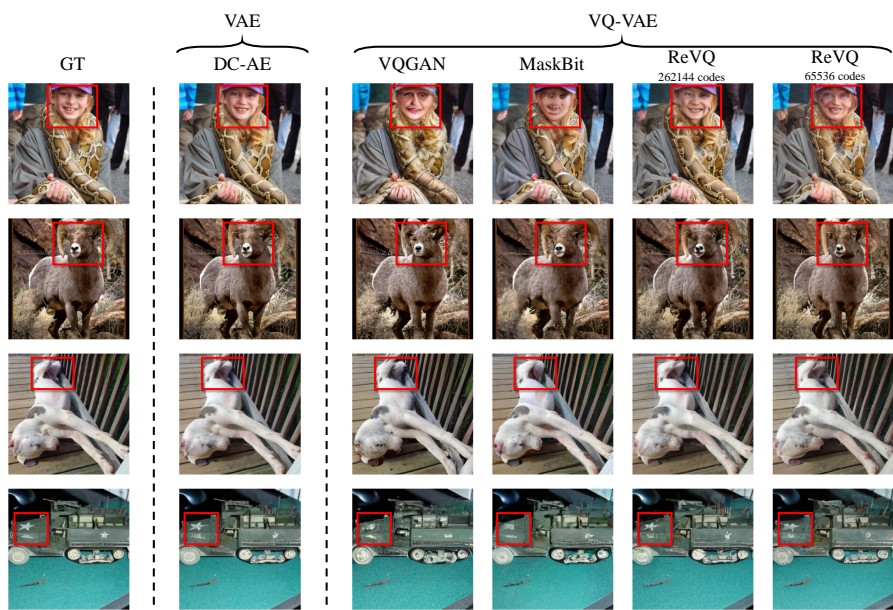

Figure 5: Reconstruction results on ImageNet validation set (details marked in red boxes).

AE (Chen et al., 2024) without further modification. ImageNet (Deng et al., 2009) images are first encoded via the VAE encoder, after which the resulting features are normalized using the global mean and variance computed from the entire dataset. These normalized features are then provided as input to our model and subsequently compressed into tokens of either 512 or 256 dimensions. The model is optimized by minimizing the $\ell_2$ loss between the normalized latent features and their reconstructions. The quantizer incorporates a **channel split** strategy in conjunction with a **non-activation reset** strategy. The codebook size is 16384 for 512 token length, while it is set to 65536 or 262144 for 256 token length. Unless otherwise specified, we implement the rectifier using a three-layer EfficientViT (Cai et al., 2023) block for 512 token length and a four-layer EfficientViT block for 256 token length. Both configurations maintain consistent input and output dimensions.

**Optimizer Setting.** All models are implemented using the PyTorch (Paszke et al., 2019) framework and trained on a single NVIDIA 4090 GPU. AdamW (Loshchilov and Hutter, 2019) is used as the optimizer. The batch size is determined dynamically based on the codebook size and GPU memory constraints. Specifically, when the token length is configured as 512, a batch size of 128 is employed. For a token length of 256, a batch size of 256 is applied with a codebook size of 65536, and reduced to 128 when the codebook size increases to 262144. The learning rate for all quantizers is initialized at $1.0 \times 10^{-2}$, whereas the decoder's learning rate is fixed at $5\%$ of that of the quantizer. An exponential learning rate scheduling policy is adopted. All models underwent training for 100 epochs.

### 4.2 PERFORMANCE COMPARISON

We conduct a comparative analysis of our ReVQ model against leading VQ-VAEs (Yu et al., 2022; Zheng et al., 2022; Li et al., 2025; Esser et al., 2021; Chang et al., 2022; Lee et al., 2022; Weber et al., 2024; Agarwal et al., 2025; Zhu et al., 2024; Sun et al., 2024; Yu et al., 2024b; Liu et al., 2025; Wang et al., 2025) on the ImageNet (Deng et al., 2009) dataset, evaluating on the validation set using four metrics: PSNR, SSIM (Wang et al., 2004), LPIPS (Zhang et al., 2018), and rFID (Heusel et al., 2017), as summarized in Table 1. Two salient observations emerge from our results. First, the model with a token length of 512 demonstrates superior performance across all metrics, surpassing both "Fine Tuning" and "Frozen" counterparts. Additionally, the configuration with a token length of 256 and a codebook size of 262144 achieves notable outcomes, surpassing all other 256 token length models except MaskBit (Weber et al., 2024). Second, our model exhibits a significant advantage in training efficiency. Compared with publicly available training durations of existing approaches, ReVQ reduces the total GPU hours by $40\times \sim 150\times$ in Table 2. Furthermore, Fig. 5 illustrates the visual reconstruction quality. The red-boxed regions highlight ReVQ's superior ability to preserve fine-grained details, particularly in areas involving complex textures and facial features.

Table 2: Training time across different methods.

| Method | #Code | GPUs | Training Time |
|---|---|---|---|
| MaskBit | 4,096 | 32×A100 | 3456 |
| TiTok-S-128 | 4,096 | 32×A100 | 1600 |
| ReVQ$_{512T}$ | 16,384 | 1×RTX 4090 | 22 |
| ReVQ$_{256T}$ | 65,536 | 1×RTX 4090 | 26 |
| ReVQ$_{512T}$ | 262,144 | 1×RTX 4090 | 40 |

Table 3: Spatial/channel split.

| #Token | #Code | Type | rFID |
|---|---|---|---|
| 512 | 16,384 | space | 1.11 |
| | | channel | 0.82 |
| 256 | 65,536 | space | 2.91 |
| | | channel | 2.56 |

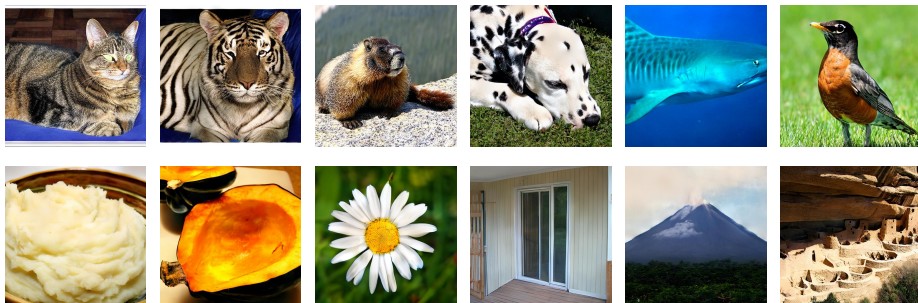

Figure 6: Generation samples on ImageNet using the ReVQ model with a diffusion model.

## 4.3 GENERATION EVALUATION

ReVQ leverages a rectifier model, serving as a decoder-only architecture, enabling seamless integration with the diffusion model employed in DC-AE (Chen et al., 2024). We first trained the ReVQ quantizer using the configuration of the corresponding diffusion model in DC-AE. Then, the ReVQ quantizer was integrated with the DC-AE decoder to evaluate its class-specific image generation performance on ImageNet (Deng et al., 2009). The generator model used is USiT-H (Chen et al., 2024), which denoises class-conditional noise in the latent space. The latent variables are then mapped to images through the ReVQ quantizer and the DC-AE decoder. As shown in Fig. 7, while ReVQ performs slightly worse than DC-AE in terms of generative performance, it still outperforms or is comparable to most existing tokenizers. Additionally, Fig. 6 showcases some of ReVQ's generation results on ImageNet. The generated images exhibit high quality and demonstrate notable diversity.

## 4.4 ABLATION STUDY ON QUANTIZER DESIGN

**Non-Activation Reset Strategy.** Nearest-neighbor-based quantizers often face the challenge of codebook collapse (Huh et al., 2023; Zhu et al., 2024; Zhang et al., 2024). To address this issue, we propose the **Non-Activation Reset** strategy in this paper. We first visualize the dynamic process of codebook changes under this strategy in Fig. 8. We randomly initialized several 2D data points,

| Method | Generator | gFID |
|---|---|---|
| Open-Magvit2-B | AR | 3.08 |
| LlamaGen-L | AR | 3.80 |
| LDM-4 | Diffusion | 3.60 |
| DC-AE | Diffusion | 1.88 |
| ReVQ$_{512T}$ | Diffusion | 2.53 |

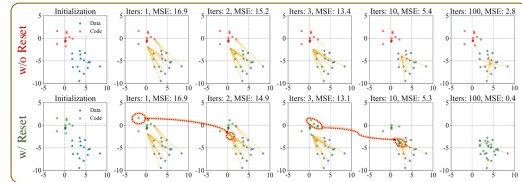

Figure 7: Quantitative comparison on ImageNet.

Figure 8: The influence of the reset strategy.

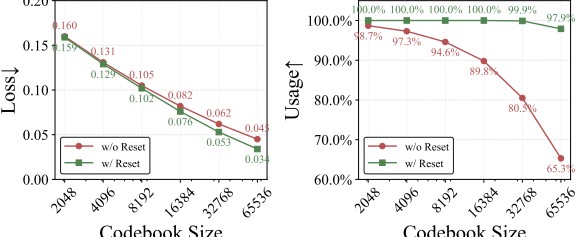

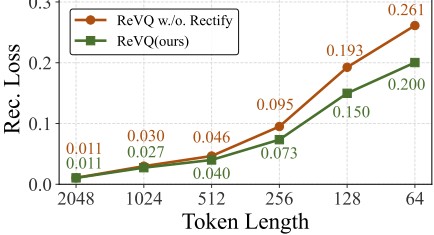

Figure 9: Ablation on reset strategy.

Figure 10: Ablation on rectifier.

Table 4: Ablations on decoder types.

| #Token | #Code | Decoder | rFID | $L_{\text{MSE}}$ |
|--------|-------|---------|------|------------------|
| 512 | 16,384 | ViT | 0.82 | 0.012 |
|  |  | CNN | 1.08 | 0.012 |
|  |  | MLP | 1.09 | 0.013 |
| 256 | 65,536 | ViT | 2.56 | 0.064 |
|  |  | CNN | 3.00 | 0.068 |
|  |  | MLP | 4.58 | 0.076 |

Table 5: Exploration of whether to add encoder.

| #Token | #Code | with Encoder? | rFID | $L_{\text{MSE}}$ |
|--------|-------|---------------|------|------------------|
| 512 | 16,384 | ✔ | 189.10 | 0.620 |
|  |  | ✗ | 0.82 | 0.012 |
| 256 | 65,536 | ✔ | 200.46 | 0.653 |
|  |  | ✗ | 2.56 | 0.064 |

each represented by 1 token. Without the reset strategy, the codebook is heavily influenced by the initialization, resulting in only a few codes being used (e.g., only 2 codes in this case) and a quantization error of 2.8. With the Reset strategy, inactive codes are reset to data-dense regions during training, as shown by the orange dashed arrows in the figure. This ensures all codes are used, reducing the quantization error to 0.4. To more thoroughly demonstrate the effectiveness of this strategy, we conducted quantitative experiments on 10% of the ImageNet dataset, as shown in Fig. 9. The results show that without the reset strategy, codebook utilization decreases rapidly as the codebook size increases, with only 65.3% of the codes utilized. In contrast, with the reset strategy, codebook utilization remains above 97% without significant decline as the codebook size increases.

### 4.5 ABLATION STUDY ON RECTIFIER DESIGN

**Effectiveness of Rectification.** We initiate our analysis by evaluating the impact of the rectifier module on model performance in Fig. 10. We conduct training on the ImageNet dataset using different token lengths and their corresponding codebook sizes, with consistent training strategies and an identical rectifier design. The use of the rectifier consistently reduces reconstruction loss across all token lengths. Notably, the improvement is more pronounced when the baseline model is weaker. Specifically, at a token length of 64, the rectifier yields a 23.3% decrease in reconstruction loss, highlighting its effectiveness in improving representational fidelity under constrained settings.

**Diverse Rectifier Architectures.** We investigate the impact of different rectifier architectures (ViT, CNN and MLP) on model performance. Experiments on the ImageNet dataset are conducted under two settings: one with a token length of 512 and codebook size 16384, and another with token length 256 and codebook size 65536, keeping all other settings identical. Empirical results demonstrate that the ViT rectifier outperforms CNN and MLP across both configurations. Conventional VQ-VAEs use symmetrical architectures (Agarwal et al., 2025; Weber et al., 2024; Yu et al., 2024b). As mentioned in Fig. 4a, a decoder-only structure is more conducive to optimization. Building on this, we explored adding an encoder matching the rectifier architecture in Table 5, but this significantly increased training difficulty and rFID. Therefore, we opted not to include an extra encoder before the quantizer.

## 5 DISCUSSION

This paper addresses the issue of time-consuming training in conventional VQ-VAEs. We discover that a pre-trained continuous feature autoencoder (VAE) and a discrete feature VQ-VAE exhibit an inherent connection. If the quantization error generated by the quantizer is smaller than the tolerance threshold, the VAE model can be seamlessly converted into a VQ-VAE model. Specifically, we propose a strategy named **Quantize-then-Rectify**. First, we freeze the parameters of the pre-trained VAE and directly apply a **channel split quantization** strategy to transform continuous features into discrete tokens. During training, we introduce a simple **non-activation reset** strategy to address the commonly encountered "codebook collapse" problem. To further reduce quantization errors, a learnable ViT model is employed as a **rectifier** after the quantizer to correct the quantized tokens. Our experiments on the ImageNet dataset demonstrate that the proposed **ReVQ** method can achieve a VQ-VAE with high compression ratio after approximately 1 day of training on a single 4090 GPU server. In contrast, conventional VQ-VAE methods requiring comparable performance necessitate 4.5 days of training on a 32 A100 GPUs. However, as shown in Fig. 10, ReVQ currently cannot match state-of-the-art approaches like TiTok in achieving extremely high compression ratios (e.g., compressing images into 32 tokens). We attribute this limitation to the architectural design of the rectifier and plan to explore more reasonable model designs in future work to enhance the compression capability of ReVQ. Beyond this, we will investigate the applicability of ReVQ across more data modalities (such as video reconstruction) and downstream tasks (such as image generation), aiming to broaden the methodological scope of ReVQ.

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

## TABLE OF CONTENT FOR APPENDIX

## A   USAGE OF LARGE LANGUAGE MODELS IN PAPER WRITING

In the process of writing this paper, large language models (LLMs) were utilized in the following two aspects:

**Preliminary Literature Review**: During the literature review phase, we used LLMs to quickly familiarize ourselves with the background knowledge and recent advancements in the relevant field. We first conducted a preliminary survey using the LLM, followed by further manual searches for related papers on arXiv and Google Scholar to verify the information.

**Assistance in Paper Writing**: During the writing process, we employed LLMs to help check for grammatical errors, refine sentences, and generate certain descriptive content. We ensured that all generated content was thoroughly reviewed and revised to guarantee its accuracy and adherence to academic standards.

We acknowledge the potential of LLMs in academic writing but also recognize their limitations. In critical areas such as code implementation, experimental design, and result analysis, no LLMs were used. All of these parts were completed independently by our team members.

The LLMs used in the above processes were OpenAI's GPT-5. We strictly adhered to relevant usage policies and ethical guidelines to ensure the originality and academic integrity of all generated content.

## B   REPRODUCIBILITY STATEMENT

We provide the complete experimental code and details of the experimental process and settings in the supplementary material to ensure the reproducibility of this method. For specifics, please refer to the code files in the supplementary material.

## C   DETAILED ANALYSIS OF CHANNEL SPLIT QUANTIZATION

As mentioned earlier in Section 3.2.1, we employed a channel split quantization strategy to enhance the expressive power of the codebook. Specifically, for an encoded feature map $\boldsymbol{Z}_e \in \mathbb{R}^{H \times W \times D}$, merging

the first two spatial dimensions results in $\boldsymbol{Z}'_e \in \mathbb{R}^{S \times D}$, where encoding length is $B = S = H \times W$. Traditional VQ uses a single codebook $\mathcal{C}$: each spatial location's feature vector $\boldsymbol{z}^i_e = [\boldsymbol{Z}'_e]_{(\cdot,i)}$ is quantized as $\boldsymbol{z}^i_q = q(\boldsymbol{z}^i_e, \mathcal{C})$. This imposes strong symmetry inductive bias by assuming identical prior distributions $p(\boldsymbol{z}^i_e)$, limiting degrees of freedom to $N$ and causing training challenges. To tackle this issue, we introduce a **channel split** strategy: define $\boldsymbol{z}^i_e = [\boldsymbol{Z}'_e]_{(i,\cdot)}$ and perform quantization. When token length $B$ differs from feature dimension $D$, we perform secondary spatial splitting after initial channel-wise division, resulting in feature vectors of dimension $d = (H \times W \times D)/B$. Additionally, we provide a detailed comparison of the differences between channel split and space split quantization strategies in Fig. 11 and Fig. 12. The channel split quantization method improves the codebook's expressive ability by leveraging the diversity in feature dimensions. Moreover, previous ablation experiments, as shown in Table 3, have verified that the channel split quantization method outperforms the space split quantization method, achieving better reconstruction results with the former.

In our experiments, **no predefined downsampling strategy is applied**. Instead, the 64 (spatial) $\times$ 32 (channel) tensor output by the DC-AE Encoder is directly used as input to our model. Specifically, this tensor has a shape of $(8 \times 8) \times 32$, where 32 denotes the channel dimension, and $8 \times 8$ corresponds to the spatial dimensions (height and width). To accommodate varying token lengths, our core operation involves "reshaping" the tensor. Let the token length be $T$ and the channel dimension be $Dc = 32$. We consider three scenarios:

- **Case 1** ($T = Dc$): When the token length $T$ equals the channel dimension $Dc$, we "split" directly along the channel dimension, with each component quantized using an independent codebook.

- **Case 2** ($T < Dc$): When the token length $T$ is smaller than the channel dimension $Dc$, we "reshape" directly along the channel dimension, meaning some components along the channel dimension are **merged**.

- **Case 3** ($T > Dc$): When the token length $T$ exceeds the channel dimension $Dc$, we also "reshape" directly along the channel dimension; however, this implies that some components along the spatial dimension are further **split**.

For instance, with a token length of $512$, each token has a dimension of $4$, while with a token length of $256$, each token has a dimension of $8$.

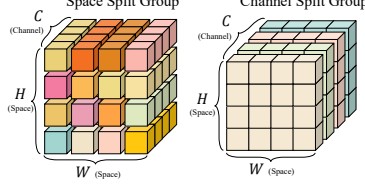

| Strategy | Split Axis | Feature Diversity |
|---|---|---|
| Space Split | Space | ✗ |
| Channel Split | Channel | ✓ |

Figure 11: Comparison of different quantizers.

Figure 12: Visualization of Different Quantization Strategies.

## D   ON THE THEORETICAL ANALYSIS OF RESET STRATEGY

We briefly analyze whether this Reset strategy can effectively reduce quantization error. Consider a code $\boldsymbol{c}_1$ activated by $m$ feature vectors $\boldsymbol{z}_i, i = 1, \cdots, m$, with its quantization error given by:

$$L_{\text{MSE}}(\boldsymbol{c}_1) = \sum_{i=1}^{m} \|\boldsymbol{z}_i - \boldsymbol{c}_1\|_2^2. \tag{5}$$

Let $\bar{\boldsymbol{z}}^{(m)}$ denote the mean of these $m$ vectors. By the least squares method, the quantization error reaches a lower bound $L_m^{(\text{lower})} \leq L_{\text{MSE}}(\boldsymbol{c}_1)$ when $\boldsymbol{c}_1 = \bar{\boldsymbol{z}}^{(m)}$. If an unactivated code $\boldsymbol{c}_2$ is reset near $\boldsymbol{c}_1$, the $m$ feature vectors are divided into two subsets $\{\boldsymbol{z}_i\}_{i=1}^{m_1}$ and $\{\boldsymbol{z}_j\}_{j=1}^{m_2}$ with $m_1 + m_2 = m$. The updated quantization error becomes:

$$L'_{\text{MSE}}(\boldsymbol{c}_1, \boldsymbol{c}_2) = \sum_{i=1}^{m_1} \|\boldsymbol{z}_i - \boldsymbol{c}_1\|_2^2 + \sum_{j=1}^{m_2} \|\boldsymbol{z}_j - \boldsymbol{c}_2\|_2^2. \tag{6}$$

Similarly, the updated lower bound $L'^{(\text{lower})}_{m_1,m_2}$ is achieved when $\boldsymbol{c}_1 = \bar{\boldsymbol{z}}^{(m_1)}$ and $\boldsymbol{c}_2 = \bar{\boldsymbol{z}}^{(m_2)}$, satisfying:

$$L'^{(\text{lower})}_{m_1,m_2} = \sum_{i=1}^{m_1} \|\boldsymbol{z}_i - \bar{\boldsymbol{z}}^{(m_1)}\|_2^2 + \sum_{j=1}^{m_2} \|\boldsymbol{z}_j - \bar{\boldsymbol{z}}^{(m_2)}\|_2^2$$

$$\leq \sum_{i=1}^{m_1} \|\boldsymbol{z}_i - \bar{\boldsymbol{z}}^{(m)}\|_2^2 + \sum_{j=1}^{m_2} \|\boldsymbol{z}_j - \bar{\boldsymbol{z}}^{(m)}\|_2^2 = L_m^{(\text{lower})}. \tag{7}$$

This analysis demonstrates that the Reset operation can effectively reduce quantization error. A 2D experiment in Section 4.4 and the detailed in Fig. 8 illustrate how the Reset operation avoids codebook collapse and thereby decreases quantization error.

## E    ALGORITHM OF REVQ METHOD

Below is the detailed algorithmic procedure for training one epoch using the ReVQ method.

---

**Algorithm 1** One-epoch training algorithm for ReVQ.

---

**Input:** Set of latent feature maps $\mathbb{Z}_e$, quantizer $q$, rectifier $g$.
**Output:** Optimized quantizer $q$ and rectifier $g$.
  1: **for** each $\boldsymbol{Z}_e \in \mathbb{Z}_e$ **do**
  2:     Quantize $\boldsymbol{Z}_e$ to obtain $\boldsymbol{Z}_q = q(\boldsymbol{Z}_e, \mathcal{C})$ via channel split strategy in Section 3.2.1.
  3:     Rectify $\boldsymbol{Z}_q$ to produce $\boldsymbol{Z}'_q = g(\boldsymbol{Z}_q)$ via the rectifier model defined in Section 3.3.
  4:     Calculate the loss function $L_{\text{ReVQ}}$ as specified in Eq. (4).
  5:     Perform backpropagation to update the parameters of rectifier $g$ and codebook $\mathcal{C}$.
  6: **end for**
  7: Apply Non-activation Reset to codebook $\mathcal{C}$ as defined in Eq. (2).
  8: **return**  Quantizer $q$ and rectifier $g$.

---

## F    IMPLEMENTATION DETAILS

### F.1    DATASETS

This study was primarily conducted on the ImageNet dataset (Deng et al., 2009). The training set of ImageNet comprises 1281167 images, while the validation set contains 50000 images, both divided into 1000 classes. To enhance the training efficiency of the ReVQ model, we first employed the DC-AE model to encode all training images into 2048-dimensional vectors. The website for the ImageNet dataset is: `http://www.image-net.org/`.

### F.2    CONFIGURATIONS

We did not employ any special data augmentation methods for the 2048-dimensional latent vectors. Taking our configuration with 512 tokens and a codebook size of 16384 as an example, the detailed settings for the model and the optimizer are as follows:

- Num Code: 16384.
- Num Group: 1.
- Tokens Per Data: 512.
- Decoder: dc_ae.
- In Channels: 32.
- Latent Channels: 32.
- Attention Head Dim: 32.
- Block Type: EfficientViTBlock.
- Block Out Channels: 512.
- Layers Per Block: 3.

- QKV Multiscales: [5].

- Norm Type: RMSNorm.

- Act Fn: SiLU.

- Upsample Block Type: interpolate.

- Optimizer: AdamW (Loshchilov and Hutter, 2019).

- Beta1: 0.9.

- Beta2: 0.999.

- Quantizer Weight Decay: 0.0.

- Decoder Weight Decay: 1e-4.

- Learning Rate: 1e-4.

- LR Scheduler: ExponentialLR.

- Base LR: 1e-2.

- Epoch: 100.

- BatchSize: 256.

- GPU: One NVIDIA GeForce RTX 4090.

# G  ADDITIONAL EXPERIMENTS

## G.1  RELATIONSHIP BETWEEN TOKEN LENGTH AND NUMBER OF CODEBOOKS

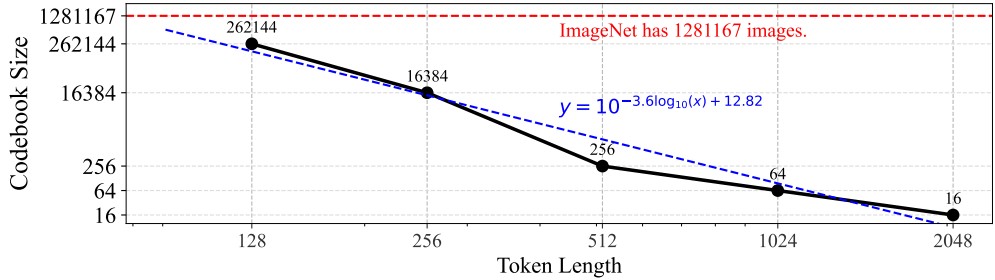

Figure 13: Relationship between the token length $B$ and the number of codebooks $M$ required to keep the quantization error below $0.1$.

We found that when the quantization error (MSE) of the latent vector is below $0.1$, the visual results of the reconstructed images are basically acceptable to the human eye. Since the dimension of the latent variable is $2048$, if the token length is $B$, the dimension of the codebook is $2048/B$. Obviously, a smaller token length $B$ leads to a higher codebook dimension. To ensure the quantization error is below $0.1$, a larger number of codebooks is required. Therefore, we explored the relationship between the token length $B$ and the minimum number of codebooks $M$ needed to keep the quantization error below $0.1$ in Fig. 13. Specifically, only the quantizer was used in this experiment without employing the rectifier to further correct the quantization error. It can be observed that the minimum number of codebooks $M$ and the token length $B$ exhibit an exponential relationship. When the token length is less than $256$, the minimum number of codebooks $M$ increases rapidly, approaching the sample size of ImageNet (1281167 images). We fitted this curve and obtained the approximate relationship:

$$M \approx 10^{-3.6 \log_{10} B + 12.82}. \tag{8}$$

Based on this, we conclude that directly using the quantizer to quantize the latent vector of a trained VAE model has obvious performance limitations. Only by introducing additional nonlinear modules can the blue curve in Fig. 13 be shifted downward to achieve higher compression rates, which is a goal we hope to further pursue at the conclusion of this work.

## G.2 DETAILED RESULTS DURING TRAINING

To further demonstrate the effectiveness of the proposed ReVQ model, we present the overall loss curves recorded during training. As shown in Figs. 14a and 14b (where the reconstruction error of quantizer features is abbreviated as "Qua Loss" and the reconstruction error of rectifier features is termed "Dec Loss" in the figures), the quantizer loss remains nearly unchanged, which is expected given that only the rectifier structure is varied while all other components are held constant. In contrast, the decoder loss is significantly affected by the rectifier type, with the ViT counterpart achieving the lowest loss. Figs. 14c and 14d further illustrate that the splitting strategy has a substantial impact on both the quantizer and rectifier losses. Specifically, the channel split approach leads to consistently lower losses, indicating better overall model performance. Figs. 14e to 14h provide additional validation for the above observations. Moreover, they reveal that using 256 tokens results in higher training loss compared to the 512-token configuration, suggesting that models with fewer tokens are more challenging to train. This observation implies that, for a given VAE architecture, the achievable compression ratio has an inherent upper bound.

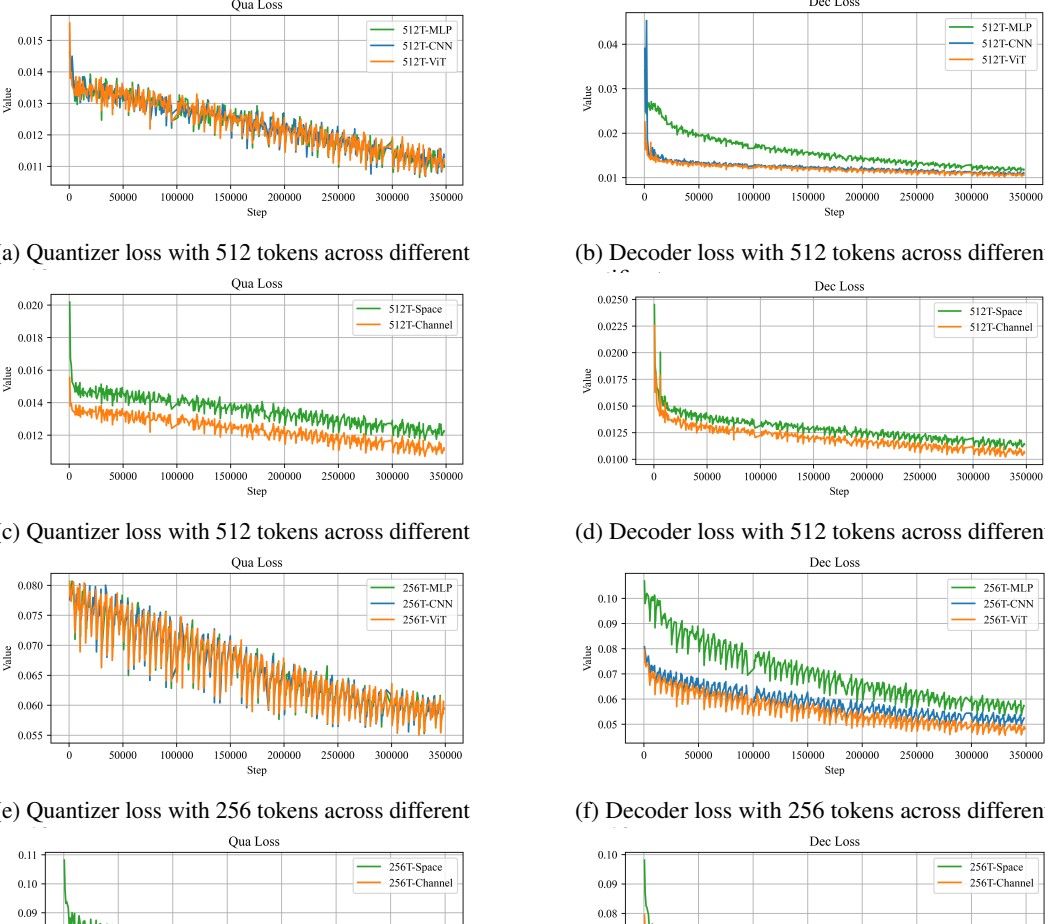

(a) Quantizer loss with 512 tokens across different

(b) Decoder loss with 512 tokens across different

(c) Quantizer loss with 512 tokens across different

(d) Decoder loss with 512 tokens across different

(e) Quantizer loss with 256 tokens across different

(f) Decoder loss with 256 tokens across different

(g) Quantizer loss with 256 tokens across different split types.

(h) Decoder loss with 256 tokens across different split types.

Figure 14: Details training statistics.

## G.3 MORE GENERATION RESULTS ON IMAGENET

In this section, we provide additional image generation results on the ImageNet dataset, as illustrated in Fig. 15.

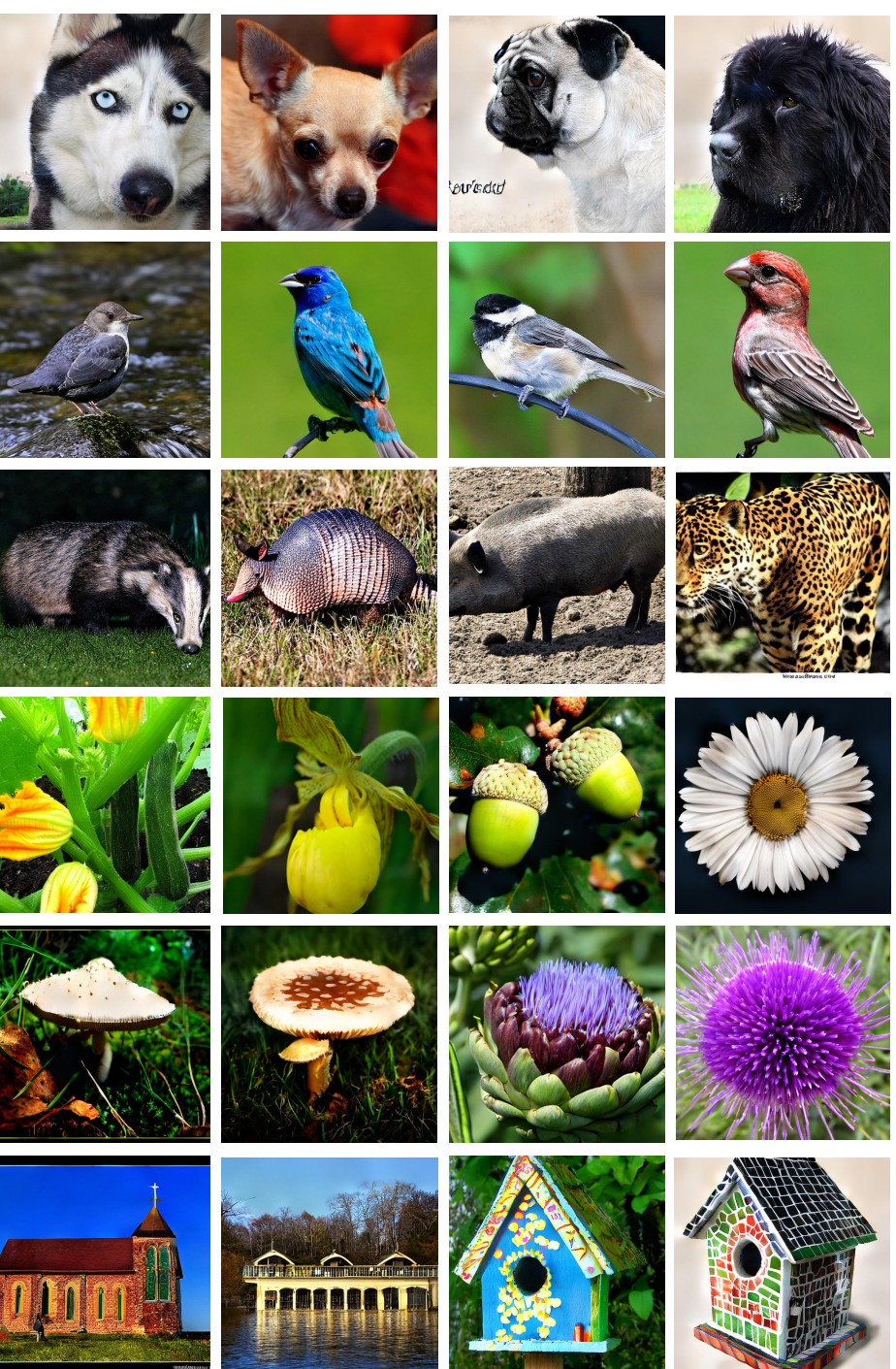

Figure 15: Additional generation results on the ImageNet dataset.

## G.4 MORE VISUALIZATIONS ON IMAGENET

In this section, we present additional reconstruction results, as shown in Fig. 16. These results further demonstrate the superior performance of the proposed ReVQ model on ImageNet dataset.

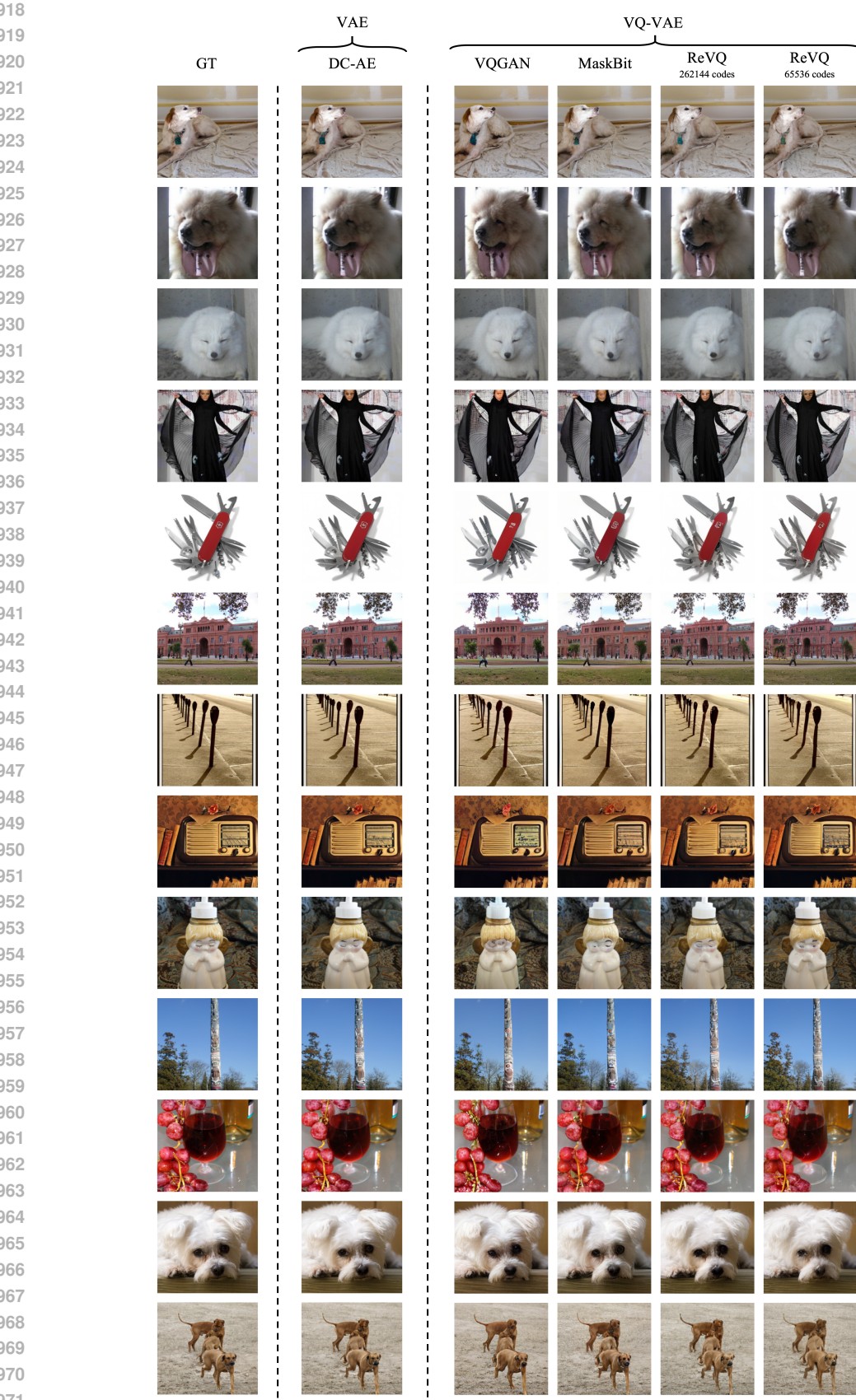

Figure 16: Additional reconstructed results on ImageNet dataset.

