# OpenReview forum: "Quantize-then-Rectify: Accelerating VQ-VAE Training in Latent Feature Space"
_ICLR.cc/2026/Conference — ICLR 2026 Conference Withdrawn Submission_

### Official Review · Reviewer_eamF · 2025-10-27

**Soundness:** 3
**Presentation:** 2
**Contribution:** 2
**Rating:** 2
**Confidence:** 5

**Summary:**

This paper introduces ReVQ, a novel framework for efficiently transforming pre-trained VAEs into high-compression VQ-VAEs for vision-language model applications. ReVQ leverages a channel split quantization and a post rectifier to overcome quantization errors. The proposed method drastically improves training efficiency.

**Strengths:**

-  ReVQ framework enables rapid training requiring only a single NVIDIA 4090 and less than a day.

- The authors conduct various experiments to show the effectiveness of the proposed results, both quantitatively and qualitatively.

**Weaknesses:**

- The authors emphasize that the ReVQ framework offers efficiency improvements. However, the core approach mainly leverages existing VAE and VQ-VAE components with only incremental modifications, rather than introducing fundamentally new architectures. The novelty of the overall approach is very limited, especially since VAE training itself is also time-consuming. It is unreasonable for the authors to ignore this part of the computation time.

- Both qualitative and quantitative evaluations are also not particularly strong. As shown in Figure 5, even with 262,144 codebook entries, the reconstruction result of ReVQ is still full of artifacts. As shown in Table 1, compared with VQGAN-LC, ReVQ has worse PSNR (22.267 vs 23.800) and LPIPS (0.122 vs 0.120) scores when using a larger codebook (262,144 vs 100,000).

- The layout and writing of the paper could be improved. The article spends a lot of space introducing traditional VQ-VAE, but put the implementation details of the proposed channel split quantization to the appendix.

**Questions:**

- If a frozen VAE combined with a learnable VQ does not outperform a VQ-VAE trained from scratch, what is the practical advantage or main contribution of your proposed approach? This question is especially important given the availability of many open-source VQ-VAE models that can be used directly. Could the authors clarify the unique benefits or scenarios where ReVQ provides significant value over existing alternatives?

- Figure 4b requires further explanation. Could the authors clarify how this figure demonstrates that channel split outperforms spatial split, and explain what specific metrics or evidence support this claim?

---

### Official Review · Reviewer_FCxv · 2025-11-01

**Soundness:** 3
**Presentation:** 3
**Contribution:** 2
**Rating:** 6
**Confidence:** 3

**Summary:**

To address the problem of high training computational cost in VQ-VAE, this paper proposes the ReVQ framework, which quickly converts a pre-trained VAE into a VQ-VAE, significantly reducing training overhead. The paper emphasizes the superiority of ReVQ in the efficiency-compression trade-off, providing an accessible visual tokenizer for large multimodal models.

**Strengths:**

The proposed ReVQ framework drastically cuts VQ-VAE training cost, greatly improving accessibility. It achieves efficiency-compression trade-off with low rFID, competitive PSNR/SSIM/LPIPS, and high-fidelity reconstruction. Extensive ImageNet evaluations and solid theoretical analysis further enhance the work’s rigor.

**Weaknesses:**

ReVQ fails to match state-of-the-art methods at extremely high compression ratios, limiting its application in extreme compression scenarios. Additionally, the experimental testing domain is narrow with insufficiently extensive evaluations, which may compromise the method's generalizability. Generalizability is not sufficiently validated. While the manuscript mentions plans to explore more modalities in the future, no evidence is provided in the current work.

**Questions:**

1. Regarding the limitation at extremely high compression ratios, how does the author plan to optimize the design of the rectifier?
2. Although the inactive reset strategy is effective, can more rigorous theoretical analysis (e.g., convergence proof) be provided?
3. How can ReVQ be extended to multimodal tasks such as video reconstruction or audio? Are there plans to validate its performance on other multimodal datasets?

---

### Official Review · Reviewer_MKFd · 2025-11-01

**Soundness:** 2
**Presentation:** 2
**Contribution:** 2
**Rating:** 2
**Confidence:** 3

**Summary:**

The paper proposes Quantize-then-Rectify (ReVQ): convert a pre-trained VAE (DC-AE) into a VQ-VAE via channel split quantization with Non-Activation Reset to improve codebook utilization, and add a decoder-only rectifier trained with ℓ2 loss on \(Z_e\) to correct quantization error.

**Strengths:**

1.  Clean separation of quantizer and rectifier \(g\); only the rectifier is trained with \(\|Z_e - g(q(Z_e))\|_2^2\), keeping optimization simple.
2. Non-Activation Reset is a direct, epoch-level heuristic to re-seed dead codes and mitigate codebook collapse under nearest-neighbor VQ.
3. Capacity diagnosis. Empirical exponential relation between token length \(B\) and the minimum number of codebooks \(M\) needed to keep MSE low clarifies compression pressure.

**Weaknesses:**

1) Noise study is not tied to the actual quantizer. The variance ≤ 0.3 tolerance is measured via Gaussian noise on DC-AE latents, not from the channel-split quantization process (its distribution, variance, or anisotropy). As a result, it does not guide codebook size or bit allocation.

2) Loss is misaligned with the evaluation target. The rectifier is trained only with latent-space ℓ2 on \(Z_e\) (decoder-only), without perceptual or adversarial terms. This weakly couples the learned correction to image-space quality, especially at high compression.

3) Scaling behavior remains unresolved. The approach is reported to compress to at most 512 tokens and implies exponential codebook growth beyond that. There is no rate–distortion analysis or entropy-regularized design to control capacity vs. fidelity at larger scales.

**Questions:**

N/A.

---

### Official Review · Reviewer_qyBb · 2025-11-05

**Soundness:** 3
**Presentation:** 3
**Contribution:** 3
**Rating:** 4
**Confidence:** 5

**Summary:**

This paper proposes a novel  framework that efficiently transforms a pre-trained VAE into a VQ-VAE by integrating channel split quantization to enhance codebook capacity and a post redtifier to mitigate quantization errors. Main results are conducted on ImageNet which achieves reducing training costs by over two orders of magnitude comparing with SOTA methods.

**Strengths:**

1. Figures are informative and clear, which are easy to follow.
2. The experimental results are appealing which obtains significant training efficiency and compression ratio improvement.
3. Motivations and corresponding methods are explained clearly.

**Weaknesses:**

1. Since pre-training a VAE also demands training time and GPU resources and previous methods do not include a pre-training stage, the proposed converting VAE into ReVQ method should add the pertaining time into consideration. Therefore, I'm curious about the total training cost including pre-training VAE stage and converting into ReVQ stage. Authors should compare the training cost and performance between VQ-VAE (trained from scratch) and ReVQ (pre-train + convert)
2. In Line 245-246, the proposed method sort each codex at the end of the epoch, what's the cost of this process and the proportion defined to determine which code can be discovered as non-activation? Authors should add relevant explanations and experimental comparison between different proportion setting about this part.
3. Although authors compare the performance between spatial and channel split strategy, there's no training cost comparison between them. Meanwhile, how about a hybrid spatial/channel split. For instance, for som sensitive channels, space split can be applied while channel split applied for the rest to achieve a trade-off between efficiency and performance.

**Questions:**

See weaknesses. I would raise my score if most of my concerns are solved (mainly the first and third point in weaknesses).

---

### Note · Authors · 2025-11-12

I have read and agree with the venue's withdrawal policy on behalf of myself and my co-authors.